# Circulating Endocannabinoids and N-Acylethanolamines in Individuals with Cannabis Use Disorder—Preliminary Findings

**DOI:** 10.3390/brainsci13101375

**Published:** 2023-09-27

**Authors:** Nadia Boachie, Erin Gaudette, Richard P. Bazinet, Lin Lin, Rachel F. Tyndale, Esmaeil Mansouri, Marilyn A. Huestis, Junchao Tong, Bernard Le Foll, Stephen J. Kish, Tony P. George, Isabelle Boileau

**Affiliations:** 1Brain Health Imaging Centre, Centre for Addiction and Mental Health, Toronto, ON N6B 1Y6, Canada; nadia.boachie@mail.utoronto.ca (N.B.);; 2Institute of Medical Science, University of Toronto, Toronto, ON M5S 1A1, Canada; 3Department of Nutritional Sciences, University of Toronto, Toronto, ON M5S 1A1, Canada; 4Department of Anatomy and Neurobiology, Faculty of Medicine, University of California, Irvine, CA 92697, USA; 5Campbell Family Mental Health Research Institute, Centre for Addiction and Mental Health, Toronto, ON N6B 1Y6, Canada; 6Department of Psychiatry, University of Toronto, Toronto, ON M5S 1A1, Canada; 7Department of Pharmacology and Toxicology, University of Toronto, Toronto, ON M5S 1A1, Canada; 8Institute of Emerging Health Professions, Thomas Jefferson University, Severna Park, Philadelphia, PA 19144, USA; 9Addictions Division and Institute of Mental Health Policy and Research, Centre for Addiction and Mental Health, Toronto, ON N6B 1Y6, Canada; 10Translational Addiction Research Laboratory, Centre for Addiction and Mental Health, Toronto, ON N6B 1Y6, Canada; 11Departments of Family and Community Medicine, University of Toronto, Toronto, ON M5S 1A1, Canada; 12Waypoint Research Institute, Waypoint Centre for Mental Health Care, Penetanguishene, ON L9M 1G3, Canada

**Keywords:** AEA, 2-AG, OEA, DHEA, NAE, N-acylethanolamines, endocannabinoids

## Abstract

Background: Endocannabinoids and related N-acylethanolamines (NAEs) are bioactive lipids with important physiological functions and putative roles in mental health and addictions. Although chronic cannabis use is associated with endocannabinoid system changes, the status of circulating endocannabinoids and related NAEs in people with cannabis use disorder (CUD) is uncertain. Methods: Eleven individuals with CUD and 54 healthy non-cannabis using control participants (HC) provided plasma for measurement by high-performance liquid chromatography–mass spectrometry of endocannabinoids (2-arachidonoylglycerol (2-AG) and N-arachidonoylethanolamine (AEA)) and related NAE fatty acids (N-docosahexaenoylethanolamine (DHEA) and N-oleoylethanolamine (OEA)). Participants were genotyped for the functional gene variant of FAAH (rs324420, C385A) which may affect concentrations of AEA as well as other NAEs (OEA, DHEA). Results: In overnight abstinent CUD, AEA, OEA and DHEA concentrations were significantly higher (31–40%; *p* < 0.05) and concentrations of the endocannabinoid 2-AG were marginally elevated (55%, *p* = 0.13) relative to HC. There were no significant correlations between endocannabinoids/NAE concentrations and cannabis analytes, self-reported cannabis use frequency or withdrawal symptoms. DHEA concentration was inversely related with marijuana craving (*r =* −0.86; *p* = 0.001). Genotype had no significant effect on plasma endocannabinoids/NAE concentrations. Conclusions: Our preliminary findings, requiring replication, might suggest that activity of the endocannabinoid system is elevated in chronic cannabis users. It is unclear whether this elevation is a compensatory response or a predating state. Studies examining endocannabinoids and NAEs during prolonged abstinence as well as the potential role of DHEA in craving are warranted.

## 1. Introduction

Cannabis is one of the most widely abused drugs globally and its use is associated with notable adverse mental health outcomes including psychosis and schizophrenia [1,2,3]. Cannabis use also increases the risk of developing cannabis use disorder (CUD) for which no approved pharmacotherapy exists [4,5,6]. There is growing concern that the legalization of cannabis use for medical and recreational purposes might increase the use of high-potency cannabis [7,8] and thus amplify the consequences of cannabis use on mental health [9].

Cannabis exerts its characteristic psychoactive effects through tetrahydrocannabinol (THC), a partial agonist at widely expressed cannabinoid receptor 1 (CB1R) [10,11,12]. The endogenous neurotransmitters for the CB1R are endocannabinoids: lipid mediators that are prevalent in the brain (and body) [13,14,15] and function as made-on-demand retrograde messengers that exert modulatory control over neurotransmitter release from axon terminals [16]. Endocannabinoids have important physiological roles in neural signaling, immune function, pain processing, cardiovascular function, homeostasis following stress, fertility, and are implicated in several diseases [17,18,19]. The two most studied endocannabinoids are the arachidonate-derived lipid molecules N-arachidonoylethanolamine (anandamide, AEA) and 2-arachidonoylglycerol (2-AG) [20,21]. 2-AG is a mono-acylglycerol lipid endocannabinoid and a full CB1R agonist; whereas AEA is a fatty acid amide that is also classified as an n-acylethanolamine (NAE) and is a partial agonist at CB1R [22]. The role of 2-AG and AEA on phasic and tonic synaptic signaling is believed to be complementary [23]. AEA is also an agonist at the transient receptor potential cation channel subfamily V member 1 receptor (TRPV1) [24] and it is broken down by the membrane-bound serine hydrolase, fatty acid amide hydrolase (FAAH); whereas 2-AG is metabolized by monoacylglycerol lipase (MAGL) [23,25,26]. The degradation pathway of 2-AG and AEA were well studied and are the focus of intense drug development [27]. For example, preclinical and clinical studies suggest that inhibiting the breakdown of NAEs (e.g., FAAH or cyclooxygenase-2 (COX-2) inhibitors) may reduce cannabis use and craving symptoms during cannabis withdrawal [28,29,30,31,32].

In addition to AEA, several other fatty acids, NAEs metabolized by FAAH, were discovered in the search for cannabinoid ligands. These include the anorexigenic lipid oleoylethanolamide (OEA) [33,34] and DHEA (N-docosahexaenoylethanolamine), (also known as synaptamide), which is known for its anti-inflammatory and neurogenic effects [35]. These NAEs are peroxisome proliferator-activated receptor-α (PPAR-α) agonists that, despite having membership in the “endocannabinoidome”, do not for the most part selectively bind CB1R at physiological concentrations [36]; although DHEA was shown to activate CB1R with significant potency in vitro [37].

Prolonged cannabis use is linked with notable adaptations in the endocannabinoid system [19,38,39]. This includes findings, from both the preclinical and clinical imaging literature, of temporary CB1 downregulation following chronic THC exposure [38,40]. We have also reported lower brain levels of the major endocannabinoid and NAE metabolizing enzyme FAAH, as measured by positron emission tomography of the radioligand [^11^C]CURB in overnight abstinent individuals with CUD. There are, however, inconsistent data on the effects of cannabis or THC on endocannabinoid and related NAEs [41]. 

In rodents, repeated exposure to THC or to the CB1/CB2 receptor agonist (WIN 55,212-2) increase plasma and brain concentrations of AEA in male and female mice [42] but the effects on 2-AG are inconsistent [40,43,44]. Similarly, in humans, studies of circulating concentrations of endocannabinoids and NAEs in regular cannabis users or following acute cannabis administration are limited and conflicting [45]. For example, acute THC administration (20 mg; Dronabinol) but not cannabis [46] administration in cannabis users, elevates concentrations of NAEs but significance is restricted to OEA [47]. In cannabis users with psychosis, both elevated [48] and decreased [49] AEA (with no change in 2-AG) is reported. Studies in cerebrospinal fluid of frequent cannabis users (defined as people who have used at least 20 times in their life) vs. controls (using cannabis < 5 times lifetime) find a trend for lower concentrations of AEA (but higher 2-AG) [50,51]. Thus far, no study has reported endocannabinoid and NAEs concentrations in cannabis users diagnosed with CUD without a co-morbidity. 

Whether or not exposure to cannabis in humans alters endogenous fatty acid (NAEs and endocannabinoids) concentration is still uncertain. A better understanding of the status of circulating endocannabinoids and NAEs and their relationship to symptomatology may help develop a better understanding of their role in CUD and inform therapeutics. The aim of this study was to investigate plasma endocannabinoid and NAE concentrations in CUD using liquid chromatography–tandem mass spectrometry in CUD and healthy controls (HC). We tentatively hypothesized, based on preclinical findings and on our earlier FAAH findings in the brain, that in CUD participants, FAAH substrates (AEA, OEA and DHEA) but not 2-AG would be elevated. 

## 2. Methods

### 2.1. Participants

All study procedures were approved by the Research Ethics Board at the Centre for Addiction and Mental Health. Some participant data were published elsewhere [41]. Participants were recruited from the local community in Toronto, Ontario, Canada through internet advertisements. After obtaining written informed consent, participants completed a comprehensive intake session to assess eligibility according to inclusion and exclusion criteria. All participants, HC and CUD, reporting past or present significant medical conditions, neurologic illnesses, head trauma, Axis I psychiatric disorders other than CUD in cannabis users and nicotine dependence in both groups, medication use that may affect the central nervous system, or positive drug toxicology for drugs of abuse other than cannabis in CUD were excluded.

Participants with CUD were required to abstain from cannabis for 12 h (overnight) before the study session, and HC and CUD were required to abstain from tobacco smoking overnight. Participants were asked not to drink caffeinated beverages on the morning of the scan. On the day of the scan, participants were given a standard meal. Overnight abstinence was verified as intake assessments included: a breath alcohol concentration measurement to ensure abstinence from alcohol; urine toxicology to rule out medication and illicit drug use (other than cannabis in CUD); a urine pregnancy test in female participants only; expired carbon monoxide (<10 ppm to rule out recent tobacco or cannabis smoking, 8–10 parts per million (ppm) for CO as a viable cut off to determine abstinence) [52]; the time line follow back to assess cannabis use over the previous 90 days [53]; and craving and withdrawal questionnaires for cannabis users. These included the severity of dependence scale, obsessive compulsive smoking scale [54], marijuana craving questionnaire-short form [55], Beck depression inventory [56] and marijuana withdrawal checklist [57]. The presence of any other psychiatric condition not assessed with these questionnaires was assessed using the structured clinical interview for DSM-IV-5 (SCID-IV-5) [58]. Biological samples for the quantification of NAE and endocannabinoids were collected once before a PET scan in the morning. PET scan methods are previously published [59]. Blood THC and metabolites quantifications were collected at two separate time points 5–6 h apart (upon arrival and at discharge) to distinguish residual from recent cannabis use [60]. 

### 2.2. Extractions of NAEs and Arachidonoylglycerols

Stock solutions of OEA, AEA, DHEA, 1, 2-arachidonoylglycerols (1- and 2-AGs) as well as [^2^H2]OEA, [^2^H8]AEA, [^2^H4]DHE, [^2^H5]2-AG were reconstituted in acetonitrile and stored at −80 °C before further dilution in acetonitrile on ice. Calibration curves were performed using various dilutions of the stock solution containing the mixture of all targeted NAE and AGs and the mixture of deuterated NAE and AGs as internal standards. Each point on the standard curve contained the same amount of internal standards as that added to the samples prior to sample processing. Results demonstrated a linear response range of 0.1–200 ng/mL for individual standards. The peak area ratios of each compound relative to the internal standard were used for calculations. The coefficient of determination (R^2^) of the calibration curve was ≥0.9910 for all fatty acid ethanolamide (FAE) species. Method accuracy and precision for each NAE were assessed in eight quality control replicates. These results indicate the high degree of reproducibility of the HPLC/MS system. All compounds were reproduced with equal to or less than 5% precision error between runs and over 95% accuracy of each targeted compounds. Specifically in this study, the accuracy range of each targeted compounds was 85–117% [61].

Solid-phase extraction was employed for human plasma samples according to previously published methods [59,61] with minor modifications. Each plasma sample was coded and labeled to maintain blinding during extraction. Distilled water (750 µL) was added to the plasma (200 µL/sample) internal standard (50 µL) mix to total 1 mL. The mixture was centrifuged at 490 g for 10 min at 4 °C. The mixture was extracted using a vacuum manifold with the Oasis HLB 1CC, 30 mg cartridge (Waters Limited, Mississauga, ON, Canada). The final eluted samples were dried under N_2_ gas and then dissolved in 100 µL of acetonitrile. All samples were stored in gas chromatography vials at −80 °C until liquid chromatography tandem–mass spectrometry (LC-MS/MS) analysis.

### 2.3. Identification and Separation Using LC-MS/MS Analysis 

Extracted human plasma NAEs and 2-AG samples were analyzed on a SCIEX QTrap5500 LC-MS/MS (Framingham, MA, USA) with an Agilent 1290 HPLC system (Agilent Technologies: Santa Clara, CA, USA). Chromatography was performed on a Phenomenex Kinetex XB-C18 column, 50 × 4.6 mm, 2.6 µm (Phenomenex, Torrance, CA, USA) at a flow rate of 600 μL/min described previously [59]. The chromatography was optimized to identify the target compounds in plasma. The 2-AG standard from Cayman contains 10% 1-AG (Appendix A). We observed conversion from 2-AG to 1-AG (Appendix A) when 2-AG standard or plasma samples were processed and stored in −80 °C before being analyzed in the LC-MS system. In our report, 2-AG and 1-AG were combined in our analysis.

### 2.4. Analysis of Cannabinoids and Metabolites

Urine and blood samples were collected from participants upon arrival at the site (T1) and before discharge (T2) approximately 5–6 h apart to measure cannabis metabolites and phytocannabinoids. Venous blood was collected in gray-topped (potassium oxalate and sodium fluoride) vacutainers. All samples were transferred to polypropylene cryotubes, frozen on dry ice, stored at –80 °C. Blood THC, 11-hydroxy-THC (11-OH-THC), THC-glucuronide (THCgluc), THCCOOH, THCCOOH-glucuronide (THCCOOH-gluc), cannabidiol (CBD), and cannabinol (CBN) concentrations were quantified by LC-MS/MS [42,62,63,64] with limits of quantification (LOQs) of 1 µg/L for THC, 11-OH-THC, THCCOOH, CBD, and CBN; 0.5 µg/L for THC-gluc; and 5 µg/L for THCCOOH-gluc. In urine, THC, 11-OH-THC, THCCOOH, CBD, and CBN were quantified by two-dimensional (2D) gas chromatography-mass spectrometry with LOQs of 2.5 µg/L for THC, 11-OH-THC, CBD, and CBN and 5 mg/L for THCCOOH [60].

### 2.5. FAAH Genotyping

All participants were genotyped for a known FAAH Single Nucleotide Polymorphism (SNP; rs324420, C385A). The FAAH genotype (rs324420) was determined using the Taqman SNP genotyping assay set performed on a ViiA7 thermal cycler (Life Technologies, Burlington, ON, Canada) with appropriate controls. Briefly, 5 μL of 2× GTXpress Master mix (cat#4401892, Life Technologies) was mixed with 10 ng of DNA and the 40 × probe (cat#C_1897306_10, Life Technologies) in a final volume of 10 μL and run for 50 cycles of 95 °C for 1 s and 60 °C for 20 s. This SNP reduces protein and steady-state activity of FAAH [65] and elevated plasma AEA concentrations [65,66,67,68].

### 2.6. Statistical Analysis

An unpaired *t*-test evaluated group differences in demographic information as well as peripheral endocannabinoids/NAE concentrations. Levene’s test for equality of variances determined whether the equal variance assumption was violated. Z-scores were used to determine outliers. Values three standard deviations above the mean were excluded from the analysis (where indicated). Absolute and percent differences in mean concentrations of peripheral endocannabinoids and NAEs were calculated between HC and CU. Pearson Chi-square tests determined if there were significant group differences in the frequencies of sex, ethnicity, FAAH polymorphism, and smoking status. Analysis of covariance (ANCOVAs) was used where necessary to control for age, sex, body mass index (BMI), smoking status and FAAH C385A genotype. Pearson product–moment correlations and Spearman’s rank–order correlations assessed the correlation between endocannabinoids/NAEs concentrations and cannabis craving, cannabis withdrawal, cannabis-use measures as well as cannabis metabolites. Partial correlations controlled for potential nuisance variables in assessing the relationship between concentrations of endocannabinoids/NAEs and craving and withdrawal measures. We also investigated with partial correlation whether endocannabinoids and NAE concentration correlated with FAAH concentrations ([^11^C]CURB binding) collected as part of our published PET study [41]. The level of significance was set at *p* < 0.05.

## 3. Results

### 3.1. Demographics

Participants’ demographic information is reported in Table 1. In total, 54 HC participants and 11 CUD participants as per SCID-IV-5 completed all aspects of the study. Demographic data from all but one cannabis user (CUD; *n* = 10) and some HC were previously reported [41]. Groups did not significantly differ in age, sex, BMI, and FAAH C385A genotype. Education tended to be slightly higher in control participants vs. cannabis users. A higher proportion of cannabis users (*n* = 6, 54.5%) reported smoking tobacco daily compared to HC (*n* = 10, 18.5%). HC and CUD who smoked cigarettes did not differ significantly with regard to number of cigarettes per day and nicotine dependence severity, as measured by the Fagerstrom test for nicotine dependence (Table 1). Cannabis users and controls did not have current or previous Axis I disorders (excluding CUD in cannabis users).

The average life-time cannabis use in the CUD group was 18 years (range: 5–33 years) with age of onset at around 16 years old (range: 10–22 years old). Cannabis users reported using an average of 7.72 g/week (range: 3.5–14) and were abstinent for approximately 21 h before the blood draws (range: 17–48 h) (Table 2). Scores from scales measuring cannabis dependence and withdrawal are reported in Table 2. 

### 3.2. Peripheral Concentrations of AEA, OEA, DHEA, and 2-AG in Cannabis Users and Healthy Controls 

Relative to HC, CUD had an elevated plasma concentration of AEA (absolute difference: 0.422 pmol/mL, 25%, *p =* 0.015), DHEA (absolute difference: 1.04 pmol/mL, 39%, *p* = 0.031), and OEA (absolute difference: 1.63 pmol/mL, 40%, *p =* 0.016). 2-AG concentrations were also elevated though non-significantly (absolute difference: 1.35 pmol/mL, 55%, *p* = 0.13) (Figure 1). One HC had plasma concentrations of AEA three standard deviations above the mean and was therefore considered an outlier. Results remain significant when this outlier was removed (AEA—absolute difference in mean: 0.502 pmol/mL, 31%, *p =* 0.001). 

Univariate ANCOVAs suggested that elevated OEA, AEA, and DHEA in cannabis users relative to controls were not accounted for by BMI, FAAH C385A genotype, and smoking status; none of these variables were significant moderators of the effect (*p* > 0.05). Males and females showed no statistically significant difference in OEA, DHEA, AEA and 2-AG in either CU or HC, or in the whole group (CUD + HC). In the whole group, age correlated with AEA (*r* = 0.45, *p* < 0.0001), OEA (*r* = 0.35, *p* = 0.004), and DHEA (*r* = 0.42, *p* = 0.001), and marginally with 2-AG (r = 0.23, *p* = 0.065) (Appendix A). This effect did not exist in cannabis users alone but was present or marginally present in HC alone (DHEA: *r* = 0.42, *p* < 0.01; OEA: *r* = 0.34, *p* = 0.13; AEA: *r* = 0.25, *p* = 0.07). In the whole group (CUD + HC), BMI correlated with 2-AG (*r* = 0.25, *p* = 0.04) (Appendix A) but not with AEA (*r* = −0.05, *p* = 0.71, OEA (*r* = −0.026, *p =* 0.84) or DHEA (*r* = 0.001, *p* = 0.991). BMI did not correlate with other measures of peripheral ECs and NAEs (*p* > 0.05) in CUD or HC alone. 

An independent samples *t*-test was conducted (without the outlier described above) to investigate whether plasma concentrations of circulating AEA, DHEA and OEA were higher in individuals with the C385A genotypes associated with lower FAAH enzymatic activity (i.e., higher in AC + AA vs. lower in CC). The test revealed that in HC, the differences were in the direction expected, but not statistically significant. OEA, DHEA and AEA were, respectively, 20% (absolute difference: 0.31 pmol/mL; *p* = 0.12), 21% (absolute difference: 0.17 pmol/mL; *p* = 0.43) and 14% (absolute difference in means: 0.04 pmol/mL; *p* = 0.18) higher in the AC + AA genotype group. Whereas in the CU, OEA, DHEA and AEA they showed no difference or showed lower concentrations of these FAAH substrates in the AC + AA genotype group (OEA—absolute difference: 0.60 pmol/mL, −10%, *p* = 0.55; DHEA- absolute difference: 0.20 pmol/mL, 5%; *p* = 0.80; AEA- absolute difference: 0.14 pmol/mL, −6%, *p* = 0.32). 

There were no significant differences between men vs. women in plasma concentrations of OEA (5.54 vs. 6.09, *p* = 0.64|4.11 vs. 4.11, *p* = 0.99), AEA (2.10 vs. 2.12, *p* = 0.87|1.76 vs. 1.62, *p* = 0.52), DHEA (4.08 vs. 2.94, *p* = 0.06|2.54 vs. 2.71, *p* = 0.79) and 2-AG (4.76 vs. 2.16, *p* = 0.07|2.54 vs. 2.39, *p* = 0.73) in CUD and HC, respectively. In a secondary analysis of healthy daily tobacco smokers (at least 1 cigarette per day) (*n* = 10) and non-smokers (*n* = 44) concentrations of OEA (47.6%; *p* = 0.08), DHEA (78.0%; *p* = 0.13), and AEA (26.1%; *p* = 0.09) (and to some extent 2-AG (15.4%; *p* = 0.48)) were non-significantly elevated in HC smokers vs. non-smokers.

In people with CUD, Pearson’s product–moment correlations were assessed relationships between clinical features of cannabis use and concentrations of peripheral endocannabinoids and NAEs. We did not find any significant correlations between the amount of cannabis used, frequency of use, number of years used, age of first use, withdrawal symptoms and peripheral concentrations of endocannabinoids and NAEs. However, there was a significant negative correlation between peripheral concentration of DHEA and marijuana craving questionnaire (MCQ) total score (*r* = −0.86; *p* < 0.01).

We did not find any significant relationship between concentrations of endocannabinoids/NAEs and concentrations of THC and metabolites (THC-gluc, THCCOOH and 11-OH-THC) and minor phytocannabinoids (CBD, CBN) (*p* > 0.05).

Data pertaining to regional FAAH concentrations in the brains ([^11^C]CURB) of people with CUD, in which we find a global 14–20% reduction in [^11^C]CURB/FAAH binding were published elsewhere [41]. We found no significant correlation between FAAH concentrations in whole brain (average brain [^11^C]CURB) and plasma concentrations of OEA, AEA, DHEA and 2-AG (*p* = 0.89, *p* = 0.48, *p* = 0.29, *p* = 0.27, respectively). 

## 4. Discussion

Our major finding is that peripheral plasma concentrations of AEA, OEA and DHEA are elevated, with a trend for increased concentrations of 2-AG in a pilot sample of individuals with CUD during early abstinence from cannabis. To our knowledge, this is the first study reporting plasma concentration of endocannabinoids and NAEs in individuals with CUD without another co-morbid psychiatric disorder. 

Endocannabinoids and related NAEs are bioactive lipids which impact multiple systems of the human body but relatively little is known about whether cannabis (i.e., THC) alters their signaling. Whereas preclinical evidence suggests that THC exposure might lead to adaptive changes in endocannabinoid signaling [69,70,71,72] and brain concentrations [29,42] only a few studies in humans reported plasma concentrations of endocannabinoids and NAEs after cannabis exposure and none were in individuals with a single diagnosis of CUD. Our preliminary results are partially in line with two of these studies. The first showed elevated AEA concentrations in individuals who abuse cannabis with a diagnosis of psychosis [48] and the second demonstrated elevated AEA, PEA and OEA in HC two hours after acute THC administration [47]. Our findings are, however, at odds with another study reporting persistently low plasma concentration of AEA and OEA in detoxified individuals with polysubstance use disorder, most of which concurrently abused cannabis [73]. The discrepancy between the later finding and our study could be due to the effects of other drugs not directly interacting with the CB_1_ receptor (i.e., stimulants and/or alcohol) or to the timing of the study (i.e., early vs. protracted abstinence).

It is unclear whether cannabis-induced alterations in endocannabinoids and NAEs are associated with central disturbances as suggested by limited preclinical studies [43,69,70]. We have previously reported a global decrease in brain concentrations of the NAE metabolizing enzyme FAAH ([^11^C]CURB) in CUD, suggesting the possibility that brain concentrations might be increased. However, despite showing here that plasma concentrations of endocannabinoids and NAEs are elevated in the same cohort, we did not find that reductions in brain FAAH correlated with elevations in peripheral endocannabinoids. While this may be an issue of insufficient power, it is nonetheless possible that different regulatory mechanisms are at play in the periphery and brain. This in fact was suggested by some studies showing that THC exposure exerts opposing regulatory effects on endocannabinoids and NAEs peripherally and centrally (i.e., feedforward vs. feedback) [47,69]. This might explain some discrepant findings in plasma vs. cerebrospinal fluid reported in the cannabis literature [48,51]. Despite this, given that endocannabinoids and NAE are highly lipophilic ligands freely diffusing across the blood–brain barrier, we cannot rule out the possibility that concentrations of NAE and endocannabinoids concentrations are also elevated in brain of individuals with CUD.

Unlike previous studies, we did not find that individuals with the FAAH C385A genotype associated with lower FAAH enzymatic activity (i.e., AC + AA) had significantly higher concentrations of AEA or NAE. Our results in HC are, however, in line with literature studies in that we find a comparable magnitude of difference. For example, our results are similar in magnitude to those of Spagnolo et al. who find a robust effect of the FAAH genotype (~15–20%) in a small sample of participants with co-morbid posttraumatic stress disorder and alcohol use disorder [74]; with those of Mayo et al. who report 19% (*p* = 0.04) higher AEA in AC + AA [75] and with those of Sipe et al. who find 11% (*p* = 0.04) higher FAAH substrates in HC participants (*n* = 144; including participants with obesity) [66]. 

The mechanism behind elevated plasma NAEs and endocannabinoids in CUD is largely undefined. While it is clear that cannabis (i.e., THC) has a direct effect on brain and body CB1/CB2 receptors leading to their downregulation (shown in post-mortem studies and PET [38,76]) and to tolerance, it is unknown whether cannabis exerts direct regulatory control over biosynthesis or release of endocannabinoids and NAEs or whether it reduces FAAH activity in brain and body (through a CB1 or PPARα mechanism) to cause their elevation. Previous preclinical studies and our own PET study [41] together with the data shown here suggest the co-occurrence of decreased FAAH and increased NAE/endocannabinoid. Without necessarily proving causality, this may indicate that elevations in endocannabinoids and NAEs could be in part be due to a downregulation (through an unknown mechanism) of FAAH in response to chronic CB1 stimulation [38]. Thus, cannabis-induced increases in NAE/endocannabinoid tone may be a homeostatic effort to restore CB_1_/CB_2_ activity, which is known to be downregulated (or the transient receptor potential cation channel subfamily V member 1 (TRPV 1) and member 2 (TRPV 2) activity). 

The clinical significance of our finding (in the periphery) is difficult to establish. It is not entirely clear how endocannabinoids are affected after acute cessation of cannabis (i.e., THC). It is unclear how cessation affects brain function and behavior. We found that AEA OEA and DHEA metabolized by FAAH and to a lesser extent, 2-AG metabolized by MAGL are elevated upon acute cannabis cessation. We also found that higher concentrations of DHEA concentrations were associated with lower self-reported craving. Our findings may suggest, on the one hand, a compensatory response to low CB1 activity. There are no studies linking DHEA with cannabis withdrawal; however, findings are in general agreement with the preclinical finding that dual FAAH/MAGL blockade produced THC-like responses that potentially mask withdrawal [29]. Importantly this is also in line with the finding that administration of the FAAH inhibitor PF-04457845 reduces symptoms of cannabis withdrawal and subsequent use [28]. On the other hand, elevated concentrations of 2-AG, a full CB1/CB2 agonist, could also perpetuate beyond THC effects, such as agonist mediated CB1 receptor desensitization, and thereby THC tolerance. Plasma endocannabinoids and NAEs concentration increases could also have effects on inflammation, metabolic functions and hypothalamic pituitary adrenal axis activity, which may be altered during acute drug cessation.

This study has several limitations. For one the sample size of the experimental group was small, yielding potentially optimistic effect sizes. Additionally, the small n was a mix of males and females, and this further reduced its power if differences exist between the sexes. Tobacco, which could influence NAE and endocannabinoid concentrations, is often co-used with cannabis [28]. Despite our effort to also include tobacco smokers in the HC sample, smoking was more prevalent in cannabis users. Additionally, factors such as nutrition, hydration and exercise frequency and intensity also influence endocannabinoid and NAE concentrations [77,78,79]. As mentioned in the methods, participants were given a standard meal in attempts to mitigate some of the short-term effects of diet. There are limitations that still exists because diet and exercise outside the parameters of the study were not monitored. The present study included individuals with high BMI, and this could add variability in the preliminary data reported [80]. BMI was taken into consideration and included in basic correlational analyses and univariate ANCOVA and showed elevated OEA, AEA, and DHEA in cannabis users, relative to controls, which were not accounted for by BMI. Despite this, a follow up study would greatly benefit from the inclusion of more participants over a broad range of BMIs. Nevertheless, our findings remained significant after statistically controlling for smoking status. Future studies should include a broader range of tobacco smokers. Factors other than cannabis use in cannabis users, for example variability in diet and sleep could affect endocannabinoid and NAE concentrations [81,82,83]. Importantly our study captures early abstinence in which it is difficult to disentangle the residual effects of cannabis from CUD. Further research should be undertaken to better capture the endocannabinoid system profile—both centrally and peripherally and longitudinally during abstinence.

There was a range of patterns of cannabis use and periods of abstinence before blood collection among the cannabis users. This may explain why endocannabinoid and NAEs did not significantly correlate with clinical features of cannabis use as measured by SDS, MWC and MCQ.

## 5. Conclusions

In conclusion, we report here, for the first time, elevated concentrations of plasma endocannabinoid and NAE together with lower self-reported craving in early abstinence in cannabis users. In light of the recent trial showing the efficacy of an inhibitor of the FAAH enzyme responsible for the metabolism of endocannabinoid in people with CUD [28], our findings suggest the need to continue characterizing endocannabinoids and NAEs and their relationship with withdrawal symptoms.

## Figures and Tables

**Figure 1 brainsci-13-01375-f001:**
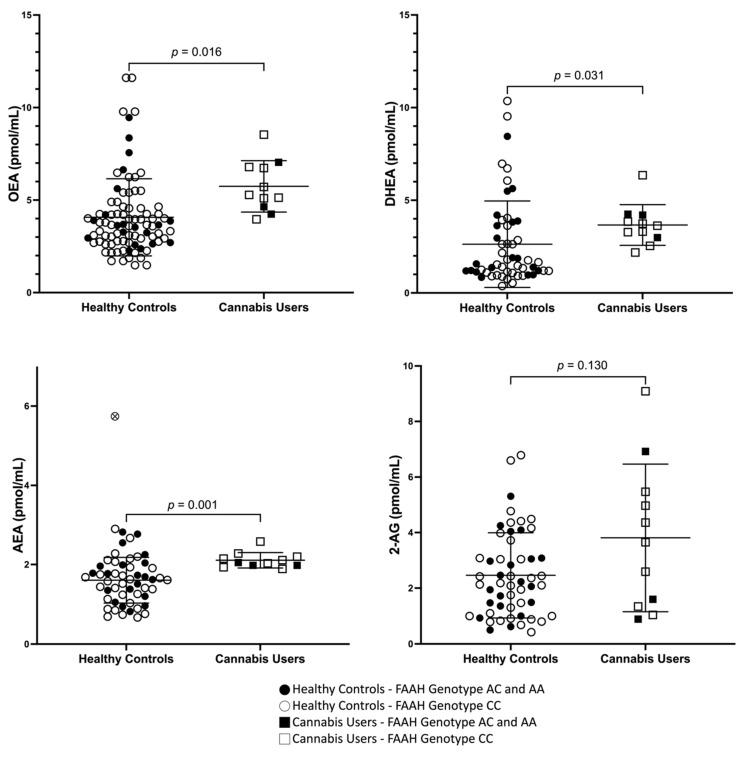
Peripheral Endocannabinoids and N-acylethanolamines in cannabis user and healthy controls. Anandamide (AEA), 2- arachidonoylglycerol (2-AG), N-docosahexaenoylethanolamine (DHEA) and N-oleoylethanolamine (OEA) in cannabis users (squares) and healthy controls (circles). One healthy control had AEA concentrations three standard deviations above the mean (white circle with cross). Removing this point resulted in a lower *p* value (*p* < 0.001). The black filled in symbols represent individuals with the FAAH C385A genotype AC and AA (black square and black circles) and open symbols represent with the CC (white square, white circle) variant. *p* values for independent samples *t*-test are given on the graph.

**Table 1 brainsci-13-01375-t001:** Demographic and Clinical Characteristics of Healthy Controls and Cannabis Users.

Characteristic	Healthy Controls (*n* = 54) Frequencies/Mean ± SD (Range)	Cannabis Users (*n* = 11) Frequencies/Mean ± SD (Range)	*p* Value x^2^
Sex (Females/Males), *n*	28/26	4/7	0.35 x^2^ = 0.88
Age, Years	28.7 ± 11.5 (19–58)	33.4 ± 9.2 (20–44)	0.21
Racial, Ethnic Categories (White/Asian/Black/Hispanic), *n*	(30/10/12/2)	(8/0/3/0)	0.56 x^2^ = 3.006
Body mass Index	24.3 ± 3.1 (18.7–32.9)	24.6 ± 4.7 (19.0–30.5)	0.79
FAAH Genotype (rs324420, C385A), *n*	34(CC), 16(AC), 4(AA)	8(CC), 2(AC), 1(AA)	0.74 x^2^ = 0.60
Education, Years	15.3 ± 2.0 (11–22)	13.9 ± 3.8 (9–19)	0.077
Tobacco Cig Smokers (>1 cigarette/day), *n*	10	6	0.01 ** x^2^ = 6.392
Tobacco Cig Per Day (in smokers)	10.6 ± 9.7 (1.0–25.0)	14.8 ± 12.5 (2–35)	0.50
Alcoholic Drinks Per Week	4.7 ± 5.7 (0–24.8)	2.5 ± 2.9 (0–7.8)	0.065
Fagerstrom Test for Nicotine Dependence,	3.6 ± 3.2 (0–8)	3.8 ± 3.3 (1–9)	0.93
Cannabis Ever Used, *n*	32	11	0.01 ** x^2^ = 6.20

Values represent mean ± SD unless otherwise indicated. ** *p* ≤ 0.01.

**Table 2 brainsci-13-01375-t002:** Cannabis Use Characteristics.

Characteristic	Mean (SD)	Range (Min–Max)
Cannabis Age of Onset, Years	15.73 (3.88)	10.00–22.00
Current Cannabis Use/Week (Grams)	7.72 (4.29)	3.50–14.00
Average Cannabis Use Joints/Week	19.83 (9.92)	2.10–35.00
Cannabis Total Years Used	18.00 (10.35)	5.00–33.0
Severity of Dependence Scale	3.27 (2.24)	0.00–7.00
Marijuana Checklist (*n* = 10)	6.00 (3.92)	1.00–12.00
Obsessive compulsive smoking scale (*n* = 10)	14.20 (7.11)	5.00–25.00
Marijuana Craving Questionnaire		
Compulsivity	1.58 (0.67)	1.00–3.00
Emotionality	3.06 (1.26)	1.00–5.30
Expectancy	4.18 (1.27)	2.00–6.30
Purposefulness	4.33 (1.56)	1.00–6.30
Total	13.15 (4.04)	5.00–19.30

## Data Availability

Not applicable.

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
