# Peer review of "Circulating Endocannabinoids and N-Acylethanolamines in Individuals with Cannabis Use Disorder—Preliminary Findings"

_brainsci, 2023, doi:10.3390/brainsci13101375_

Round 1
Reviewer 1 Report
1. The number of enrolled individuals with cannabis use disorder (CUD) is too small. This leads to limited persuasiveness.
2. Why not compare the differences in absolute quantitative data in the results section?
3. Why is it limited to observing the genetic polymorphism of FAAH? Why not consider other genes such as MAGL, DAGL, CB1R?
4. Figure 1 requires splicing specifications and there are errors.
5. How is the percentage content of each substance obtained?
Minor:
1.line322-332, line 130, line 70, font inconsistency 2.P should be “P” throughout the manuscript.Author Response
Dear Reviewer,
Re: Resubmission of manuscript reference No. brainsci-2608033
Please find attached a revised version of our manuscript originally entitled “Circulating Endocannabinoids and N-Acylethanolamines during Early Abstinence from Cannabis in Individuals with Cannabis Use Disorder”.
Thank you to the reviewer’s whose insightful comments greatly improved the clarity and quality of the manuscript. In the following pages are responses to each reviewer’s comments in a numbered list.
We hope that the revisions in the manuscript and responses make this manuscript suitable to publish in Brain Sciences.
Reviewer 1
- The number of enrolled individuals with cannabis use disorder (CUD) is too small. This leads to limited persuasiveness.
The manuscript language and title have been revised to convey that the results are preliminary. We have included language throughout the paper to indicate that a larger sample size is needed to make the findings definitive.
- Why not compare the differences in absolute quantitative data in the results section?
We have added mean values and absolute differences of means in the results section.
- Why is it limited to observing the genetic polymorphism of FAAH? Why not consider other genes such as MAGL, DAGL, CB1R?
There is a strong rational for looking at the impact of FAAH on FAAH substrates (AEA, OEA, DHEA) [1]. Investigating other SNP, while interesting was not part of the scope of the study - research ethic board approval was obtained to look at FAAH gene which has been linked with FAAH substrates.
- Figure 1 requires splicing specifications and there are errors.
The data splicing specification was given in the figure caption. We have further added a legend and have corrected inconsistencies between the text and the figure.
- How is the percentage content of each substance obtained?
We calculated % difference from healthy control values by dividing the absolute value of the difference between two numbers by the average of those two numbers. Multiplying the result by 100 will yield the solution in percent. We have mentioned the use of percent differences in Methods section.
Minor:
- Line 322-332, line 130, line 70, font inconsistency 2.P should be “P” throughout the manuscript.
This has been corrected and font is uniform throughout manuscript.
Reviewer 2 Report
This is a well written article and will be of interest to the intended audience. I have a few comments where I think that the manuscript can improve:
“We have also reported lower brain levels of the major endocannabinoid and NAE metabolizing enzyme FAAH, as measured by positron emission tomography of the radioligand [11C]CURB in overnight abstinent individuals with CUD. “ What is the citation for this work?
“All participants were genotyped for a known FAAH Single Nucleotide Polymorphism (SNP; rs324420, C385A) according to procedures outlined elsewhere” Please describe this technique in the current manuscript
I suspect that all of the references are numbered differently from within text to the reference list. I tried to find references 52 (the original study based on this data) and 60 (PET protocols) but 52 and 60 in the reference list were unrelated
Something wrong with formatting – “p values for independent samples t-test are given on the graph.” Unsure whether this belongs to Figure caption or not
BMI and age correlations – which direction do these effects go in? Ideally for these sections it would be nice to see the effects of interest (not just the significant ones) visualised in a figure
“Our finding that in CU the effect of genotype is not preserved is puzzling” – I don’t think that this is puzzling at all. I would have expected to not see a significant effect here with 8 & 3 in each group for the comparison – statistically it is very unlikely to occur.
The authors hypothesise about why endocannabinoids are higher in CUD users. The first explanation that comes to my mind is that CB1 sensitivity is downregulated with chronic cannabis use, meaning that they need higher levels of stimulation to achieve the same effect. This implies that higher levels of endocannabinoids are a compensation mechanism
Reference:
Hirvonen, J., Goodwin, R. S., Li, C. T., Terry, G. E., Zoghbi, S. S., Morse, C., Pike, V. W., Volkow, N. D., Huestis, M. A., & Innis, R. B. (2012). Reversible and regionally selective downregulation of brain cannabinoid CB1 receptors in chronic daily cannabis smokers. Mol Psychiatry, 17(6), 642-649. https://doi.org/10.1038/mp.2011.82
Declaration of interest is incomplete:
“He is supported by CAMH, Waypoint Centre for Mental Health Care, a clinician-scientist award from the department of Family and Community Medicine of the University of Toronto and a Chair in Addiction Psychiatry from the department of Psychiatry of”
Overall, I think that the abstract overplays the “early abstinence” part of the experiment. Yes, the CUD group is in an early abstinence period, but the abstract and some parts of the conclusion make it sound like higher endocannabinoid levels are caused by the abstinence rather than the chronic cannabis use. Since the current study cannot establish whether the findings are due to abstinence or cannabis use, I think that the authors should make it clearer in important sections like the abstract that the cause of the elevation is unclear (though in my opinion more likely to be due to chronic use)
Author Response
Dear Reviewer,
Re: Resubmission of manuscript reference No. brainsci-2608033
Please find attached a revised version of our manuscript originally entitled “Circulating Endocannabinoids and N-Acylethanolamines during Early Abstinence from Cannabis in Individuals with Cannabis Use Disorder”.
Thank you to the reviewer’s whose insightful comments greatly improved the clarity and quality of the manuscript. In the following pages are responses to each reviewer’s comments in a numbered list.
We hope that the revisions in the manuscript and responses make this manuscript suitable to publish in Brain Sciences.
Reviewer 2
- “We have also reported lower brain levels of the major endocannabinoid and NAE metabolizing enzyme FAAH, as measured by positron emission tomography of the radioligand [11C]CURB in overnight abstinent individuals with CUD. “What is the citation for this work?
We have added the reference.
- All participants were genotyped for a known FAAH Single Nucleotide Polymorphism (SNP; rs324420, C385A) according to procedures outlined elsewhere” Please describe this technique in the current manuscript.
We have added this to the manuscript.
- I suspect that all of the references are numbered differently from within text to the reference list. I tried to find references 52 (the original study based on this data) and 60 (PET protocols) but 52 and 60 in the reference list were unrelated.
The reference list has been updated and revised.
- Something wrong with formatting – “p values for independent samples t-test are given on the graph.” Unsure whether this belongs to Figure caption or not.
We have addressed this.
- BMI and age correlations – which direction do these effects go in? Ideally for these sections it would be nice to see the effects of interest (not just the significant ones) visualized in a figure.
We have added correlation plots (BMI and Age) in the supplementary material.
- “Our finding that in CU the effect of genotype is not preserved is puzzling” – I don’t think that this is puzzling at all. I would have expected to not see a significant effect here with 8 & 3 in each group for the comparison – statistically it is very unlikely to occur.
This statement has been removed.
- The authors hypothesize about why endocannabinoids are higher in CUD users. The first explanation that comes to my mind is that CB1 sensitivity is downregulated with chronic cannabis use, meaning that they need higher levels of stimulation to achieve the same effect. This implies that higher levels of endocannabinoids are a compensation mechanism.
We agree and do discuss this but we have clarified further. “Without necessarily proving causality, this may indicate that elevations in endocannabinoids and NAEs could in part be due to a downregulation (through an unknown mechanism) of FAAH in response to chronic CB1/ CB2 stimulation. Thus, cannabis-induced increases in NAE/endocannabinoid tone may be a homeostatic effort to restore CB1/CB2 activity which is known to be downregulated”
- Declaration of interest is incomplete: “He is supported by CAMH, Waypoint Centre for Mental Health Care, a clinician-scientist award from the department of Family and Community Medicine of the University of Toronto and a Chair in Addiction Psychiatry from the department of Psychiatry of
This has been corrected.
- Overall, I think that the abstract overplays the “early abstinence” part of the experiment. Yes, the CUD group is in an early abstinence period, but the abstract and some parts of the conclusion make it sound like higher endocannabinoid levels are caused by the abstinence rather than the chronic cannabis use. Since the current study cannot establish whether the findings are due to abstinence or cannabis use, I think that the authors should make it clearer in important sections like the abstract that the cause of the elevation is unclear (though in my opinion more likely to be due to chronic use)
We have adjusted the language in the abstract, conclusion and throughout the manuscript to reduce definitive language and emphasis on early abstinence.
Reviewer 3 Report
The authors performed an interesting study about detecting the circulating Endocannabinoids and N-Acylethanolamines during Early Abstinence from Cannabis in Individuals with Cannabis Use Disorder. There are some issues that should be discussed.
1- The statistics in table 1 need to be revised x2 in smoking is 6.392 not 0.64. The same in Ethnic categories need to be revised.
2- The sample size is small to confirm the results obtained.
3- Among the various cofounders that can affect the results of the study are nutrition, hydration, exercise, and smoking. The author tries to control smoking. What about the additional co-founders?
4- Why did the authors not compare the results of the male and female cannabinoid groups to see whether there were any gender differences?
5- The wide range of BMI reported in Table 1 that includes obese and non-obese persons could also affect the result of the current study as the body fat affects the serum level of Endocannabinoids and N-Acylethanolamines.
Author Response
Dear Reviewer,
Re: Resubmission of manuscript reference No. brainsci-2608033
Please find attached a revised version of our manuscript originally entitled “Circulating Endocannabinoids and N-Acylethanolamines during Early Abstinence from Cannabis in Individuals with Cannabis Use Disorder”.
Thank you to the reviewer’s whose insightful comments greatly improved the clarity and quality of the manuscript. In the following pages are responses to each reviewer’s comments in a numbered list.
We hope that the revisions in the manuscript and responses make this manuscript suitable to publish in Brain Sciences.
Reviewer 3
- The statistics in table 1 need to be revised x2 in smoking is 6.392 not 0.64. The same in Ethnic categories need to be revised.
We have corrected this.
- The sample size is small to confirm the results obtained.
Thank you for this concern. We have addressed this issue by altering language to indicate that the findings are preliminary.
- Among the various cofounders that can affect the results of the study are nutrition, hydration, exercise, and smoking. The author tries to control smoking. What about the additional co-founders?
Participants receive a standard meal on the day of the scan to minimize this. This is added in the limitation section of the discussion and included in the methods. We have also listed other limitations and confounds like BMI, age and diet and have added additional graphs in supplementary material. We have also included more notable mentions of these confounds in the manuscript.
- Why did the authors not compare the results of the male and female cannabinoid groups to see whether there were any gender differences?
We have looked at differences in Endocannabinoids and N-Acylethanolamines across biological sex but found no statistical significance (reported in manuscript).
- The wide range of BMI reported in Table 1 that includes obese and non-obese persons could also affect the result of the current study as the body fat affects the serum level of Endocannabinoids and N-Acylethanolamines.
We have reported correlations of BMI and Endocannabinoids and N-Acylethanolamines and included supplementary graph of statistically significant results. In the limitation section we have also acknowledge that the group included people with high BMI and that could add to variability in preliminary data.
[1] V. Di Marzo, and Mauro Maccarrone. , "FAAH and anandamide: is 2-AG really the odd one out?." Trends in pharmacological sciences vol. 29.5, pp. 229-233., 2008.
Round 2
Reviewer 1 Report
I don't have any more suggestions.
Reviewer 2 Report
No further comments
Reviewer 3 Report
The authors address my concerns